# Multiparametric Approach to the Colorectal Cancer Phenotypes Integrating Morphofunctional Assessment and Computer Tomography

**DOI:** 10.3390/cancers16203493

**Published:** 2024-10-15

**Authors:** Patricia Guirado-Peláez, Rocío Fernández-Jiménez, Francisco José Sánchez-Torralvo, Fernanda Mucarzel Suárez-Arana, Fiorella Ximena Palmas-Candia, Isabel Vegas-Aguilar, María del Mar Amaya-Campos, Gema Martínez Tamés, Virginia Soria-Utrilla, Francisco Tinahones-Madueño, José Manuel García-Almeida, Rosa Burgos-Peláez, Gabriel Olveira

**Affiliations:** 1Department of Endocrinology and Nutrition, Virgen de la Victoria University Hospital, 29010 Málaga, Spain; pguirado1991@gmail.com (P.G.-P.); rociofernandeznutricion@gmail.com (R.F.-J.); isabel.mva13@gmail.com (I.V.-A.); mariadelmarac2@gmail.com (M.d.M.A.-C.); fjtinahones@hotmail.com (F.T.-M.); 2Instituto de Investigación Biomédica de Málaga y Plataforma en Nanomedicina-IBIMA Plataforma BIONAND, 29010 Málaga, Spain; fjstsol@gmail.com (F.J.S.-T.); virginiasoriau@gmail.com (V.S.-U.); gabrielolveiracasa@gmail.com (G.O.); 3Department of Medicine and Dermatology, Málaga University, 29016 Malaga, Spain; 4Department of Endocrinology and Nutrition, Quironsalud Málaga Hospital, Av. Imperio Argentina, 29004 Malaga, Spain; 5Unidad de Gestión Clínica de Endocrinología y Nutrición, Hospital Regional Universitario de Málaga, 29007 Málaga, Spain; 6Endocrinology and Nutrition Department, Hospital Universitari Vall D’Hebron, 08035 Barcelona, Spain; fernanda.mucarzel@vhir.org (F.M.S.-A.); rosa.burgos@vallhebron.cat (R.B.-P.); 7Diabetes and Metabolism Research Unit, Vall d’Hebron Institut De Recerca (VHIR), 08035 Barcelona, Spain; 8Department of Medicine, Universitat Autònoma de Barcelona, 08193 Barcelona, Spain; 9Department of Endocrinology and Nutrition, Valle del Nalon Hospital, 33920 Asturias, Spain; gemartinez@outlook.es; 10Department of Endocrinology and Nutrition, Hospital Universitario Virgen de la Victoria, CIBEROBN, Carlos III Health Institute (ISCIII), University of Málaga, 29016 Malaga, Spain

**Keywords:** CT imaging, colorectal cancer, body composition, morphofunctional assessment, cancer patients

## Abstract

Patients with colorectal cancer (CRC) have a high prevalence of malnutrition, which is associated with a decrease in overall survival. In this study, we analyzed body composition using techniques such as BIVA, NU, HGS, and CT in a population with CRC and overweight and found that there is a high prevalence of sarcopenia in these patients. Early detection and treatment of sarcopenia will be essential to improving the prognosis of these patients.

## 1. Introduction

Colorectal cancer (CCR) is a major global health challenge, being the third most commonly diagnosed malignancy and the second leading cause of cancer-related deaths worldwide [1]. Efforts to reduce mortality rates have focused on early detection and improving the quality of life by reducing cancer-related complications. However, most CCR patients experience significant side effects that lead to decreased energy intake and severe nutritional problems, including loss of body weight and muscle mass, which adversely affect overall survival [2]. The prevalence of malnutrition, sarcopenia, and cachexia among CCR patients undergoing chemotherapy and radiotherapy is high, independent of body mass index (BMI), and these conditions are negatively associated with survival and increased mortality [3,4,5].

Traditional anthropometric measurements and BMI have been used to predict malnutrition and sarcopenia [6], but these metrics can be confounded by factors such as water retention and altered muscle-to-fat ratios, especially in cancer patients [7,8,9]. Detailed assessments of body composition are therefore crucial to reveal true metabolic changes, including muscle quality and fat distribution [10,11]. Tools such as standardized phase angle by BIVA or global subjective global assessment (SGA) have been associated with postoperative complications and overall survival in patients with RCC [12], suggesting that comprehensive nutritional assessment is critical to identifying patients at increased risk and implementing timely interventions [13].

Phase angle (PhA) from bioelectrical impedance analysis (BIVA) has emerged as a significant tool for assessing cellular health and body composition [6]. PhA is indicative not only of nutritional status but also of overall mortality in various pathologies [7,8]. In cancer patients, PhA has been shown to correlate with overall survival and the risk of complications [9]. For instance, studies have demonstrated that PhA is associated with sarcopenia and can predict cancer complications with high accuracy [10]. These findings underscore the importance of PhA as a predictive marker for sarcopenia and complications in CCR patients, potentially guiding nutritional interventions that could improve survival outcomes [11,12].

In addition, computed tomography (CT) has become increasingly important in the assessment of body composition in CRC patients. CT is highly accurate and reliable in diagnosing sarcopenia and has been linked to prognosis after colorectal surgery, as it provides information on both muscle quantity and quality through Hounsfield Units (HU). Despite being a more expensive technique, its widespread use in the staging of CRC patients allows us to make opportunistic use of this technique; moreover, there is a wide range of specific software available for body composition analysis. Computed tomography (CT) scans are routinely utilized in nearly all colorectal cancer (CRC) patients prior to intervention and have become increasingly valuable in the assessment of body composition in these patients as an opportunistic measure [13]. CT is highly accurate and reliable for diagnosing sarcopenia and has been correlated with prognosis following colorectal surgery. Despite its widespread use in clinical practice, the high cost of CT and the requirement for specialized software and expertise limit its applicability, particularly in resource-constrained settings.

Nutritional ultrasound, especially the assessment of the rectus femoris cross-sectional area (RF-CSA), has emerged as a promising technique for evaluating muscle mass, quantity, and quality through the grayscale and detecting sarcopenia [14]. Ultrasound is non-invasive, cost-effective, and portable, making it an attractive option for routine clinical use. Studies have shown that ultrasound measurements of muscle mass are a viable alternative for assessing body composition in clinical settings, providing information on complications and sarcopenia in different pathologies [15,16,17,18,19]. For instance, in the DRECO study, ultrasound cut-off values for detecting sarcopenia in hospitalized patients at risk of malnutrition were established [20], further validating the technique’s clinical utility.

The aim of our study is to integrate morphofunctional assessment techniques, such as Bioelectrical Impedance Vector Analysis (BIVA), Nutritional Ultrasound (NU), functional tests, and CT scans, to enhance body composition assessment in clinical practice. While CT scans remain the gold standard for evaluating body composition, the inclusion of these additional morphofunctional techniques can provide valuable insights, particularly in scenarios where CT is not feasible or accessible. This integrated approach has the potential to increase the accuracy of nutritional assessments and improve the management and prognosis of patients with colorectal cancer (CRC).

## 2. Materials and Methods

### 2.1. Study Design

This cross-sectional study included 267 patients recruited from University Hospital “Virgen de la Victoria” (Málaga, Spain) and Vall d’Hebron (Barcelona, Spain) from October 2019 to June 2023. Inclusion criteria encompassed outpatient individuals with a diagnosis of colorectal neoplasia who were attended to at the Nutrition Unit before colorectal surgery. Exclusion criteria excluded subjects who had undergone their last CT scan more than three months prior to the consultation or patients hospitalized at the time of assessment. Following the nutritional assessment, patients are provided with nutritional recommendations, exercise, and nutritional supplementation with immunonutrition. All patients provided written informed consent. The study was reviewed and approved by the Ethics Committee of Vall d’hebron University Hospital on 29 February 2024 (reference number #PR(AG)489/2021). The patient selection process is illustrated in a flow chart diagram (Appendix A).

### 2.2. Anthropometric and Morphofunctional Assessment

In this study, a comprehensive morphofunctional assessment was conducted on CCR patients. This included not only a combination of advanced body composition techniques to evaluate muscle mass, fat distribution, and overall nutritional status (GLIM criteria) but also hand strength and functional tests. The assessment comprised the following:

#### 2.2.1. BIVA

Body composition measurements were performed with a 50 kHz phase-sensitive impedance analyzer (BIVA 101 Whole Body Bioimpedance Vector Analyzer, AKERN, Florence, Italy), delivering 800 μA via tetrapolar electrodes positioned on the right hand and foot. All BIVA determinations were obtained with the subject supine in a hospital bed. To establish BIVA values [±2 Ω for Resistance (R) and ±1 Ω for Xc (Reactance)], the subject lay supine for five minutes before BIVA measurements were recorded, as fluid changes occurred after the change from standing to recumbency and directly affected R and Z measurements. The subjects’ BIVA values were standardized for gender and age using data from healthy Italian adults. BP is given in degrees as arctan (Xc/R) × (180o/π). An individually standardized Pha value (SPA) was calculated from the reference population value paired by sex and age by subtracting the reference BP value from the observed subject’s BP value and dividing the result by the reference standard deviation (SD) of the respective sex and age. The technical accuracy of the BIVA instrument was evaluated daily using a precision track supplied by the manufacturer of the BIVA device (AKERN, Florence, Italy). All measured values of R and Xc were consistently ±1 Ω of the reference value of 385 Ohm. The in vivo reproducibility of BIVA measurements was also determined, with coefficients of variation (CV) of 1–2% for R and Xc. We utilized the Piccoli software [21] to plot the individual bioimpedance vectors of our sample. These vectors were represented within the updated tolerance ellipses by gender, corresponding to the bivariate 50th, 75th, and 95th percentiles of the new reference healthy population [22].

#### 2.2.2. Nutritional Ultrasound

Rectus femoris quadriceps muscle ultrasonography (RF-CSA) of the lower extremity was performed with a 10–12 MHz probe and a multifrequency linear array (Mindray Z60, Madrid, Spain) in all subjects (subject in supine position). The evaluation was carried out without any compression at the level of the lower third from the superior pole of the kneecap and the anterosuperior iliac spinous, with measurement of anteroposterior muscle thickness, circumference, and cross-sectional area. The ultrasound was carried out by a specific clinician who had previously trained in this procedure. The probe was positioned perpendicular to the longitudinal and transverse axes on the QRF, such as the rectus femoris cross-sectional area (RF-CSA), rectus femoris circumference (RF-CIR), RF axis (-X and -Y axes), and L-SAT (subcutaneous fat of the leg). Three measurements were performed for each parameter, and the mean was then calculated. For adipose tissue assessment, in the abdomen, the middle point between the xiphoid appendix and the umbilicus was measured for imaging, where T-SAT (total subcutaneous abdominal fat), S-SAT (superficial adipose subcutaneous abdominal fat), and VAT (preperitoneal or visceral fat) data will be measured in centimeters.

#### 2.2.3. Functional Assessment

Hand grip strength (HGS) is measured with a JAMAR hand dynamometer (Asimow Engineering Co., Los Angeles, CA, USA). HGS was measured in a sitting position with the elbow flexed to 90 degrees in the dominating arm. The patients were asked to execute three maximal isometric contractures with short breaks between the measurements, and the mean value was registered.

#### 2.2.4. CT FocusedOn^®^

To obtain the area of skeletal muscle and abdominal adipose tissue, cross-sectional CT images with a slice at the third lumbar vertebra (L3) were examined using FocusedON-BC software (version 1.0). This software has a user-friendly interface and includes a semi-automatic model labeling tool that allows the user to automatically adjust the body mass orientation performed by the software. Previously obtained diagnostic CTs of the patients were used, provided that they had been obtained no more than 3 months prior to the assessment at the nutrition clinic. The muscles included in the analyses were psoas, erector spinae, quadratus lumborum, transversus abdominis, external and internal obliques, and rectus abdominis. Adipose tissue was assessed and classified as subcutaneous, visceral, and intramuscular. Skeletal muscle area or SMA (cm^2^ and %), skeletal muscle index or SMI (cm^2^/m^2^), intramuscular adipose tissue area or IMAT (cm^2^ and %), and mean Hounsfield units (HU) were recorded for each tissue section. Abdominal computed tomography (CT) images focused on the L3 vertebrae were obtained using a multidetector computed tomography scanner. Previously obtained diagnostic CTs of the patients were used, provided that they had been obtained no more than 3 months prior to the assessment at the nutrition clinic. The muscles included in the analyses were psoas, erector spinae, quadratus lumborum, transversus abdominis, external and internal obliques, and rectus abdominis. Adipose tissue was assessed and classified as subcutaneous, visceral, and intramuscular.

The following variables were recorded: skeletal muscle mass area or SMA (cm^2^ and %), skeletal muscle mass index or SMI (cm^2^/m^2^), intramuscular adipose tissue area or IMAT (cm^2^ and %), intramuscular adipose tissue index or IIMAT (cm^2^/m^2^), area of visceral fat mass (VFA) (cm^2^ and %), subcutaneous fat (SFA) (cm^2^ and %), visceral fat mass index (VFI) (cm^2^/m^2^), subcutaneous fat (SFI) (cm^2^/m^2^), and mean Hounsfield Units (HU) for each segmented tissue. The CT images centered on the third lumbar vertebra (L3) were analyzed using FocusedON-BC software. Tissue density was assessed based on its average Hounsfield Units (HU) value. Standard thresholds were used as follows: −29 to 150 HU for skeletal muscle, −190 to −30 for subcutaneous adipose tissue, and −150 to −50 for visceral adipose tissue [23,24,25].

### 2.3. Assessment of Sarcopenia and Low Muscle Mass

A comparison with previous literature was made to determine whether the sample values in our study were within the appropriate range for CT-L3 measurements. Specifically, we compared the Skeletal Muscle Index (SMI) cut-off points for men and women from Prado et al. [26] and Martin et al. [27] to determine if the values in our sample aligned with these established benchmarks. To ensure our SMI measurements from CT scans were comparable with previous publications, we combined the muscle SMA and IMAT, adjusted by height ((SMA + IMAT)/height^2^). This approach was necessary as Martin et al. did not differentiate between IMAT and SMA tissue. According to Prado et al., the SMI cut-off points are ≤52.4 cm^2^/m^2^ for men and ≤38.5 cm^2^/m^2^ for women with a BMI ≥30. According to Martin et al., the SMI cut-off points for men are <53 cm^2^/m^2^ for a BMI ≥25 and <43 cm^2^/m^2^ for a BMI <25, while for women, it is <41 cm^2^/m^2^ regardless of BMI.

We also used the cut-off points for subcutaneous adipose tissue (SAT-CT) from Caan et al. [28], which set the threshold at >270 cm^2^ for women and >203 cm^2^ for men. For the assessment of myosteatosis, we referred to Dolan et al. [29], who provided cut-off values of <34 Hounsfield Units (HU) for individuals with a BMI <25 kg/m^2^ and <32 HU for those with a BMI ≥25 kg/m^2^, applicable to both men and women.

Sarcopenia was diagnosed according to the EWGSOP2 criteria. Probable sarcopenia was indicated by a handgrip strength (HGS) of <16 kg for women and <27 kg for men. When low muscle mass was also present, defined as appendicular skeletal muscle mass (ASMM) <20 kg for men and <15 kg for women, or appendicular skeletal muscle mass index (ASMMI) <5.5 kg/m^2^ for women and <7.0 kg/m^2^ for men (measured by BIVA) the diagnosis of sarcopenia was confirmed.

### 2.4. Statistical Analyses

The data analysis was performed using the JAMOVI program (version 2.3.28 macOS). Continuous variables were presented as mean ± standard deviation (SD), while categorical variables were presented as numbers and percentages. An exploratory analysis was conducted to understand the data distribution and relationships. Depending on the normality of the variables, either a Student’s *t*-test or Wilcoxon test was used for comparisons. Pearson correlation coefficients were calculated and represented in a heatmap, complemented by a Cronbach’s analysis to assess internal consistency. Linear regression analyses were also performed to evaluate the predictive power of various factors affecting CT-SMI in patients with colorectal cancer (CCR). A significance level of *p* < 0.05 was considered for all two-tailed tests.

## 3. Results

As shown in Table 1, the study cohort consisted of 267 patients with an average age of 68.2 years. The sample included 61.8% males and 38.2% females, with an average BMI of 26.8 kg/m^2^. Colon cancer was the most common diagnosis, representing 80.5% of cases, while rectal cancer accounted for 19.5%. At the time of evaluation, 23.2% of patients were in stage I, 31.5% in stage II, 34.1% in stage III, and 5.6% had an unknown stage. The majority of surgeries (95.1%) were performed laparoscopically, with only 4.9% conducted as open surgeries. In our cohort, no significant differences were observed between males and females in terms of age, BMI, exitus, or cancer type distribution. However, a significant difference was observed in the length of hospital stay.

### 3.1. Body Composition Parameters and Functional Status: BIVA, NU, HGS, and CT

There are significant differences between males and females in terms of muscle mass and fat distribution, as shown in Table 2. The analysis shows a notable trend of higher muscle mass-related parameters in males compared to females. Specifically, males have significantly higher values for phase angle, body cell mass (BCM), fat-free mass (FFM), fat-free mass index (FFMI), appendicular skeletal muscle mass (ASMM), and appendicular skeletal muscle mass index (ASMMI). The majority of the sample bioimpedance vectors by gender were plotted to the right of the major axis and below the minor axis, indicating higher body fluid accumulation and lower muscle mass (Figure 1).

In addition, rectus femoris cross-sectional area (RF-CSA) and HGS are significantly greater in men. In contrast, females have a higher percentage of fat mass (FM%) and increased measurements of leg subcutaneous adipose tissue (L-SAT), abdominal subcutaneous adipose tissue (A-SAT), and preperitoneal adipose tissue (A-VAT). These differences are statistically significant, with *p*-values below 0.001 for most parameters, indicating greater fat distribution in women.

Table 3 shows the different body composition profiles between males and females according to the CT parameters. Males have significantly higher muscle area and muscle percentage, as well as muscle density, measured by Hounsfield units (HU) and skeletal muscle index (SMMI). Intramuscular adipose tissue area and percentage (IMAT) show no significant differences between genders, but visceral adipose tissue area (CT-VAT) is significantly higher in males. In contrast, females have a significantly higher subcutaneous adipose tissue area (SAT).

### 3.2. Comparison with Reference Value of Sarcopenia

The assessment of sarcopenia in CCR patients was conducted using multiple criteria, summarized in Table 4. The table highlights the prevalence of sarcopenia using CT-based European Working Group on Sarcopenia in Older People (EWGSOP2) criteria, as well as SMI cut-off points from Martin et al. [27] and Prado et al. [26] Based on EWGSOP2, nine patients (3.7%) were diagnosed with sarcopenia. For handgrip strength, 70 patients (27.5%) had low values, and regarding appendicular skeletal muscle mass index (ASMMI), 27 patients (11.2%) had low ASMMI. According to Martin’s criteria, 43.8% of patients had low SMI, and according to Prado’s criteria, 49.8% had low SMI.

The evaluation of high-fat mass and muscle quality was carried out using CT measurements, with the results detailed in Table 5. This table includes reference values for subcutaneous adipose tissue (SAT) and muscle quality based on Hounsfield Units (HU). According to the cut-off points established by Caan et al. [28] for SAT-CT, 77 patients (28.8%) were identified as having high-fat mass. In terms of muscle quality, which was assessed as myosteatosis using criteria from Dolan et al. [29], 65 patients (24.3%) were found to have poor muscle quality.

### 3.3. Correlation Analysis between Muscle Measures: CT, BIVA, NU, and Functional Test (HGS)

The heatmap in Figure 2 shows the correlation coefficients between different muscle tissue parameters. The skeletal muscle index CT-SMI has a strong positive correlation with body cell mass (BCM) (r = 0.65, *p* < 0.001) and a moderate correlation with phase angle (PhA) (r = 0.6, *p* < 0.001). In addition, CT-SMI correlates with handgrip strength (HGS) (r = 0.55, *p* < 0.001) and rectus femoris cross-sectional area (RF_CSA) measured by nutritional ultrasound (NU) (r = 0.56, *p* < 0.001). Muscle cross-sectional area (muscle_cm^2^) also shows strong positive correlations with BCM (r = 0.78, *p* < 0.001), PhA (r = 0.57, *p* < 0.001), HGS (r = 0.72, *p* < 0.001), and RF_CSA (r = 0.63, *p* < 0.001). Muscle HU (muscle density measured in Hounsfield units), hand grip strength (HGS), and phase angle (PhA) suggest a moderately positive correlation between muscle density and cell health or quality, with correlation coefficients of Cronbach’s alpha 0.789.

### 3.4. Correlation Analysis between Adipose Measures: CT, BIVA, and NU

The heatmap in Figure 3 shows the analysis of correlation coefficients, highlighting the relationships between CT-based measurements and various adipose tissue and anthropometric variables. CT-SAT (subcutaneous adipose tissue) shows positive correlations with fat mass measured by bioimpedance (FM) (r = 0.81, *p* < 0.001), BMI (r = 0.78, *p* < 0.001), A-SAT (r = 0.67, *p* < 0.001), and L-SAT (r = 0.65, *p* < 0.001) measured by nutritional ultrasound (NU). The Cronbach’s alpha value of 0.527 indicates that the current set of parameters is less internally consistent than the muscle parameters but shows a certain degree of consistency.

### 3.5. Regression Model Results

The regression analysis demonstrates the robust predictive power of the model for CT-SMI in colorectal cancer (CCR) patients, with an R-value of 0.892 and an R^2^ value of 0.796, explaining approximately 80% of the variance in CT-SMI. By incorporating diverse assessment techniques—gender, age, current weight, rectus femoris cross-sectional area (RF-CSA), handgrip strength (HGS), and body cell mass (BCM)—the model effectively captures the complexity of muscle mass evaluation (Table 6).

## 4. Discussion

The results presented in this study confirm that the different morphofunctional techniques, such as BIVA, NU, HGS, and CT, have a strong correlation when measuring the different muscle and adipose compartments. Furthermore, it has been shown that multiparametric analysis of the evaluation of muscle mass using different techniques can improve the accuracy of the diagnosis and personalized intervention of the different therapeutic options [10,30].

Traditional methods for evaluation of nutritional status, such as BMI and anthropometric measurements, are often inadequate for patients with colorectal cancer (CRC) due to their inability to accurately reflect changes in body composition, muscle quality, and fat distribution [31]. These parameters may be artifactualized by factors such as water retention and altered muscle-to-fat ratios. Detailed evaluations of body composition are, therefore, essential to understanding metabolic changes in patients with CRC and to guide nutritional interventions [32].

To describe our sample of patients with colon cancer, the composition has been analyzed by different body composition techniques and compared with each other for correlation.

BIVA-derived phase angle (PhA) has emerged as an important marker for the evaluation of cellular health, body composition, and overall mortality in cancer patients [33]. Studies have shown that body cell mass (BCM) correlates with sarcopenia, muscle quality, and risk of complications in patients with CCR [18,19]. In our study, the mean BCM in men was 34.4 kg and in women 25.9 kg, as demonstrated in the study by Prior et al., describing that higher BCM was significantly associated with a lower risk of mortality (OR, 0.88; 95% CI, 0.80–0.96; *p* < 0.01) and a lower likelihood of hospitalization (OR, 0.91; 95% CI, 0.83–0.99; *p* < 0.05) [9,10,34]. Both males and females in our sample were located in the lower right quadrant of the tolerance ellipses, a common position in patients with lower muscle mass and a higher risk of malnutrition, cachexia, or sarcopenia [35,36]. Furthermore, in the study by Espinosa et al., BIVA represents a reliable alternative to DEXA for measuring body composition, which is considered the gold standard due to the good agreement of the Bland-Altman plot [37].

The results of our study are consistent with the existing literature, which highlights similar sex-based differences in body composition in cancer patients. For example, Mortellaro et al. observed that muscle values are generally higher in men than in women, correlating with higher PhA and hand grip strength (HGS) values in men. These observations are consistent with our results, where males showed higher muscle mass parameters than females. Furthermore, the review by Mortellaro et al. highlights the limitations of traditional methods, such as BMI, for the evaluation of body composition in cancer patients. Our study also showed that, although BMI values were similar in men and women, there were significant differences in muscle mass and fat distribution, underscoring the inadequacy of BMI as an independent measure of nutritional status in patients with colorectal cancer (CCR). Our results, together with those of Mortellaro et al., support the integration of advanced body composition evaluation techniques such as BIVA and computed tomography (CT) into routine clinical practice [38].

In addition, a recent study by de Luis et al. [22] also shows significant differences in BIVA and NU values classified by sex, higher in men, as in our CCR sample with similar RF-CSA and *Y*-axis values that determine sarcopenia and malnutrition.

On the other hand, if we focus on the results of adipose tissue measured by different techniques, women have higher fat values measured by BIVA (FM), NU (SAT), and CT (SAT). However, higher VAT values were found in men when measured by CT, which is corroborated by the results of Dolan et al. [29], where VAT areas were >160 cm^2^ in men and >80 cm^2^ in women. In our sample, the mean IVA values were 200 cm^2^ in men and 143 cm^2^ in women, showing significant differences between sexes.

In our study, we used previously published cut-off points to compare the risk of sarcopenia, excess fat, or myosteatosis. To determine sarcopenia according to CT, we used the cut-off points of Prado et al. [27] and Martin et al. [30]. In our sample of patients with RCC, 49.8% had low muscle mass using Prado’s cut-off point and 43.8% using Martin’s, which differs from the data published by Prado et al. (15% sarcopenia). However, the prevalence of sarcopenia evaluated by other techniques using the EWGSOP2 criteria in our sample is lower (3.7%). This discrepancy suggests that CT cut-off points may overestimate the prevalence of sarcopenia, indicating the need for new cut-off points that better reflect the reality of patients with colorectal cancer. The main difference between the two study samples is that in our sample, the mean BMI was 27 kg/m2 (overweight), and the cut-off points of Prado et al. were established according to mortality.

For subcutaneous adipose mass (SAT), we used the cut-off values of Caan et al. [28], observing a 43% elevated SAT-CT, similar to our results of 50.9%. These values suggest a significant accumulation of subcutaneous adipose tissue, which is relevant for the evaluation of metabolic risk and body composition in patients with CCR.

Myosteatosis was evaluated by muscle density in Hounsfield units (HU), with cut-off values according to Dolan et al. [29], highlighting the importance of both muscle quantity and quality as prognostic factors in patients with CCR. Their results showed 60.3% of myosteatosis, compared to 24.4% in our sample, with a slightly higher prevalence in women (13.5%) compared to men (10.9%). Myosteatosis in our sample could be underestimated as focused on offers the fat infiltrating the muscle as separate data from the muscle area, so muscle HU is underestimated [38].

Correlation analysis of our study shows that detailed body composition evaluations using BIVA and NU are comparable to CT scans [12,17,18]. In addition to providing information about the morphology of the muscle, CT also provides information about the quality and, therefore, could correspond to functionality. In our sample, a strong correlation between muscle area measured on CT with HGS and muscle HU has been demonstrated, as also described in the study by Mortellaro et al. [39].

The value of CT muscle mass analysis in explaining body composition measurement can be analyzed through a linear regression model where the role of basic components, such as age, sex, and weight, together with advanced body composition evaluation parameters such as RF-CSA, BCM, and HGS are evaluated. In the 2022 publication by Wei Ji [40], the original regression model used weight, height, and sex as predictors, achieving an adjusted R^2^ of 0.597. This suggests that 59.7% of the variability in BMI could be explained by these basic parameters. Our alternative model incorporated additional predictors, including BCM, age, weight, RF-CSA, and HGS, resulting in an improved R^2^ of 0.796, which explains almost 80% of the variability in muscle area measured by CT.

Among the strengths of this study is the integration of multiple body composition evaluation techniques, providing a more comprehensive and detailed view of the nutritional and functional status of patients with CCR [41]. In addition, this study provides valuable data on the prevalence and factors associated with sarcopenia and obesity in patients with CCR, contributing to the existing literature and providing a solid basis for future research and clinical applications.

However, there are some limitations to this study. Firstly, its observational design does not allow causal relationships to be identified. In addition, the study sample, although representative, is limited to two hospitals in Spain, which may affect the applicability of the results to other populations. Studies should compare the efficacy of these techniques in detecting sarcopenia and myosteatosis with established methods such as CT, especially in longitudinal studies evaluating patient outcomes. In addition, investigating the impact of early detection and personalized interventions based on CT measurements on postoperative recovery and survival in patients with colorectal cancer could provide valuable information.

## 5. Conclusions

This study highlights the importance of integrating advanced body composition assessment techniques, such as BIVA, NU, HGS, and CT, in the evaluation of patients with colorectal cancer (CRC). Our findings reveal significant correlations between BIVA, NU, HGS, and CT measurements, highlighting their complementary roles in the assessment of muscle mass, fat distribution, and overall nutritional status. In cases where CT is not available or bedside assessment of body composition is required, it can be concluded that HGS, BIVA, and NU are able to accurately assess body composition, leading to targeted interventions, better treatment, and potentially a better prognosis for CRC patients.

## Figures and Tables

**Figure 1 cancers-16-03493-f001:**
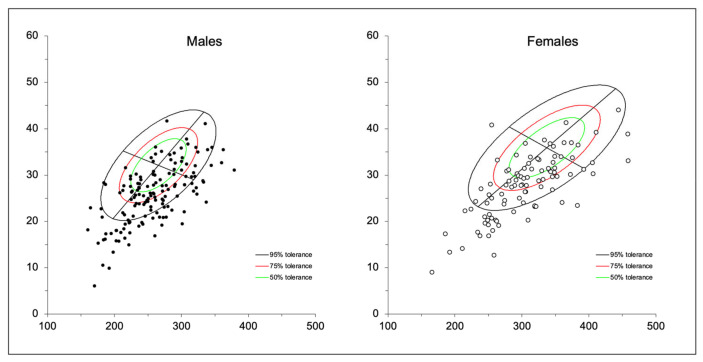
Individual bioimpedance vectors of CCR patients plotted on the updated tolerance ellipses by gender.

**Figure 2 cancers-16-03493-f002:**
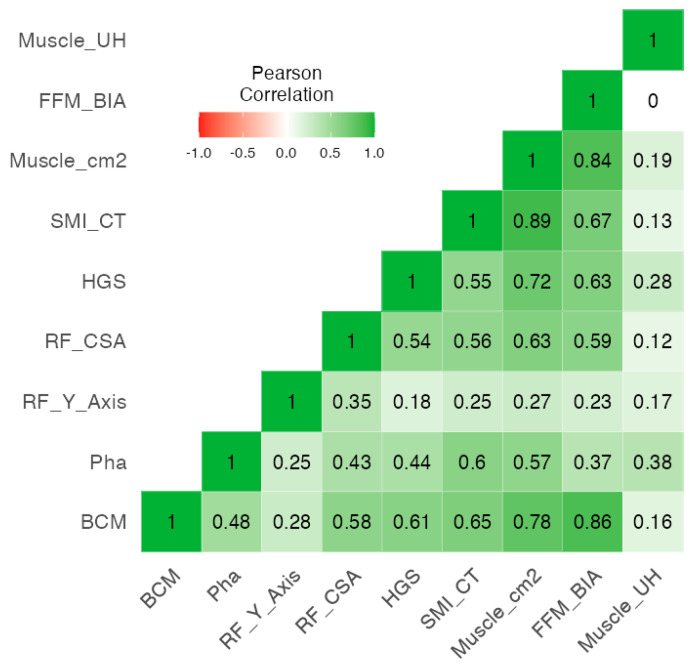
Correlation heatmap between muscle tissue by different techniques.

**Figure 3 cancers-16-03493-f003:**
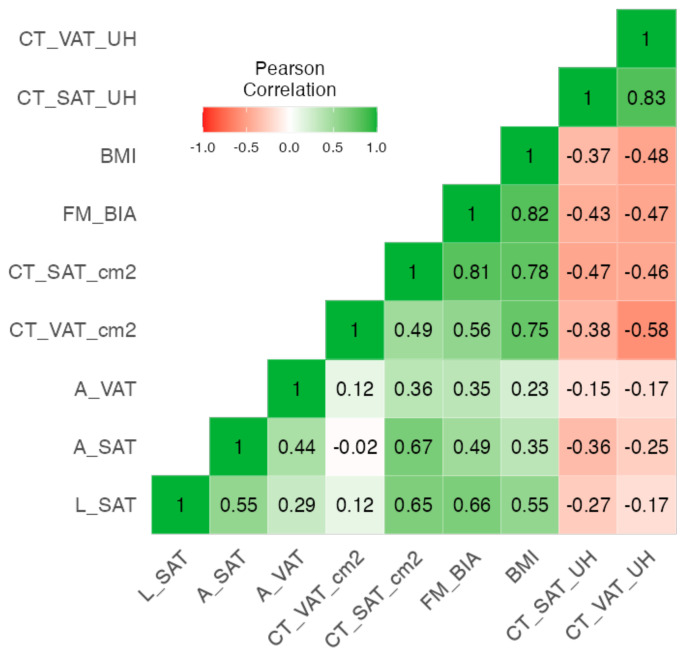
Correlation heatmap between adipose tissue by different techniques.

**Table 1 cancers-16-03493-t001:** Comparison in demographic and clinical characteristics between genders at CCR patients.

	All	Male	Female	*p*-Value
	*N = 267*	*N = 165*	*N = 102*	
**Age (years)**	68.2 ± 10.9	68.3 ± 11.4	68.1 ± 9.97	0.87
**Gender**		165 (61.8%)	102 (38.2%)	
**BMI (kg/m^2^)**	26.8 ± 4.93	26.5 ± 4.30	27.3 ± 5.80	0.28
**Malnutrition GLIM criteria** **Type of cancer**	99 (37.1%)	61 (22.8%)	38 (14.2%)	0.96
Colon	215 (80.5%)	131 (49.1%)	84 (31.5%)	
Rectum	52(19.5%)	34 (12.7%)	18 (6.7%)	
**Stage**				
Unknown at valuation	15 (5.6%)	8 (3.0%)	7 (2.6%)	0.57
I	62 (23.2%)	37 (13.9%)	25 (9.4%)	
II	84 (31.5%)	56 (21.0%)	28 (10.5%)	
III	91 (34.1%)	54 (20.2%)	37 (13.9%)	
IV	15 (5.6%)	10 (3.7%)	5 (1.9%)	
**Type of surgery**				0.57
Open	13 (4.9%)	9 (3.4%)	4 (1.5%)	
Laparoscopic	254 (95.1%)	156 (58.4%)	98 (36.7%)	
**Outcomes**				
Days of admission	7.32 ± 6.43	8.02 ± 7.48	6.19 ± 3.99	0.02 *
Éxitus	23 (8.6%)	17 (6.4%)	6 (2.2%)	0.21
Immediate Complication	65 (24.3%)	43 (15.7%)	23 (8.6%)	0.59

Data are expressed as mean ± standard deviations or percentages. Groups were divided by gender variable according to the median value. Asterisk indicates a significant difference between groups, according to the Mann–Whitney test (Chi-squared test was used for variables expressed as percentages (* *p* < 0.05). Abbreviations: BMI = body mass index.

**Table 2 cancers-16-03493-t002:** Body composition parameters and functional status.

	Male (n = 165)	Female (n = 102)	*p*-Value
**BIVA** ** Raw Bioelectrical data **			
Rz	439 ± 77.1	483 ± 89.1	<0.001
Xc	43.9 ± 10.7	43.9 ± 10.5	0.955
Phase angle (°)	5.71 ± 1.15	5.20 ± 0.927	<0.001
BCM (kg)	34.4 ± 6.68	25.9 ± 5.06	<0.001
** Validate BIVA equation **			
FFM (kg)	59.8 ± 9.07	45.4 ± 7.29	<0.001
FFMI (kg/m^2^)	20.6 ± 2.81	18.0 ± 2.34	<0.001
FM (kg)	18.8 ± 9.12	20.9 ± 9.24	0.080
FM (%)	24.4 ± 11.3	30.3 ± 10.9	<0.001
ASMM (kg)	22.8 ± 3.48	17.2 ± 3.14	<0.001
ASMMI (kg/m^2^)	7.89 ± 1.06	6.83 ± 1.16	<0.001
**Nutritional Ultrasound (NU)**			
Rectus femoris cross-sectional area (RF-CSA) (cm^2^)	4.35 ± 1.45	3.22 ± 1.03	<0.001
RF-X axis (cm)	3.82 ± 0.508	3.74 ± 3.06	0.731
RF-Y axis (cm)	1.36 ± 0.348	1.24 ± 1.04	0.174
Leg Subcutaneous adipose tissue (L-SAT) (cm)	0.594 ± 0.293	1.49 ± 1.90	<0.001
Abdominal Subcutaneous adipose tissue (A-SAT) (cm)	1.31 ± 0.643	2.19 ± 0.899	<0.001
Preperitoneal adipose tissue (A-VAT) (cm)	0.685 ± 0.295	0.939 ± 0.418	0.001
**Functional test**			
Handgrip strength (kg)	32.7 ± 9.06	19.1 ± 6.64	<0.001

Abbreviations: FFM: fat-free mass; FFMI: fat-free mass index; BIVA: bioelectrical impedance analysis; ASMM: appendicular skeletal muscle mass; ASMMI: appendicular skeletal muscle mass index; VAT: visceral adipose tissue; SAT: subcutaneous adipose tissue.

**Table 3 cancers-16-03493-t003:** Body composition parameters by CT.

		Male N = 165	Female N = 102	*p*-Value
Muscle area (SMA)	mean ± SD	130 ± 23.7	92.4 ±15.8	<0.001
Muscle (%)	mean ± SD	18.0 ± 4.31	14.3 ± 4.21	<0.001
Muscle (HU)	mean ± SD	41.0 ± 9.31	38.1 ± 9.86	0.015
SMI-CT	mean ± SD	44.8 ± 7.47	36.8 ± 5.60	<0.001
IMAT area	mean ± SD	15.7 ± 11.4	17.3 ± 10.6	0.256
IMAT (%)	mean ± SD	2.65 ± 4.28	2.48 ± 1.27	0.703
IMAT (HU)	mean ± SD	−64.4 ± 6.33	−65.8 ± 6.74	0.095
CT-VAT area	mean ± SD	200 ± 117	143 ± 83.3	<0.001
CT-VAT (%)	mean ± SD	31.0 ± 37.9	19.4 ± 8.76	0.003
CT-VAT (UH)	mean ± SD	−93.8 ± 8.58	−93.3 ± 9.18	0.650
CT-SAT area	mean ± SD	156 ± 70.4	232 ± 124	<0.001
CT-SAT (%)	mean ± SD	26.3 ± 41.3	31.8 ± 10.5	0.190
CT-SAT (HU)	mean ± SD	−96.6 ± 11.5	−99.8 ± 11.3	0.027

Abbreviations: SD: standard deviation; HU: Hounsfield units; SMI: skeletal muscle index; IMAT: intra-muscular adipose tissue; VAT: visceral adipose tissue; SAT: subcutaneous adipose tissue.

**Table 4 cancers-16-03493-t004:** Reference value of sarcopenia criteria.

Reference Value	Total	n = 267
**Sarcopenia CT**		
**SMI** (kg/m^2^)		
Low SMI (Martin)	Total	117 (43.8%)
Male	n (%)	75 (28.1%)
Female	n (%)	42 (15.7%)
Low SMI (Prado)	Total	133 (49.8%)
Male	n (%)	109 (40.8%)
Female	n (%)	24 (9%)
**Sarcopenia (EWGSOP2 criteria)**	n (%)	9 (3.7%)
**Handgrip strength**		
Low HGS	Total	60 (27.5%)
Male	n (%)	39 (15.3%)
Female	n (%)	31 (12.2%)
**ASMMI** (kg)		
Low ASMMI	Total	27 (11.2%)
Male	n (%)	22 (9.1%)
Female	n (%)	5 (2.1%)

Abbreviations: CT: computed tomography; SMI: skeletal muscle index; ASMMI: appendicular skeletal muscle mass index.

**Table 5 cancers-16-03493-t005:** Reference value of Hight fat mass CT.

SAT-CT cm^2^	Total	n = 267
**Hight fat mass CT** (Caan)	Total	77 (28.8%)
Male	n (%)	41 (15.4%)
Female	n (%)	36 (13.5%)
**Muscle Quality (UH)**		
**Myoesteatosis CT** (Dolan)	Total	65 (24.3%)
Male	n (%)	29 (10.9%)
Female	n (%)	36 (13.5%)

**Abbreviations**: CT: computed tomography; HU: Hounsfield Units; SAT: subcutaneous Adipose Tissue.

**Table 6 cancers-16-03493-t006:** Model coefficients CT-SMI in patients with CCR.

			95% Confidence Interval			
Predictor	Estimate	SE	Lower	Upper	t	*p*
Intercept	23.211	23.7882	−23.681	70.102	0.976	0.330
Gender:						
Male Female	14.821	2.6860	9.526	20.116	5.518	<0.001
Age	−0.332	0.0919	−0.513	−0.151	−3.618	<0.001
Weight	0.547	0.0811	0.387	0.707	6.743	<0.001
Hight	9.834	14.6346	−19.014	38.682	0.672	0.502
RF_CSA	2.298	0.8618	0.600	3.997	2.667	0.008
HGS	0.524	0.1428	0.242	0.805	3.668	<0.001
BCM	0.808	0.1944	0.425	1.191	4.155	<0.001

Abbreviations: BCM: body cell mass; RF-CSA: rectus femoris cross-sectional area; HGS: hand grip strength.

## Data Availability

Data sharing is not applicable. No new data were created or analyzed in this study.

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
