# Peer review of "Multiparametric Approach to the Colorectal Cancer Phenotypes Integrating Morphofunctional Assessment and Computer Tomography"

_cancers, 2024, doi:10.3390/cancers16203493_

Round 1
Reviewer 1 Report
Comments and Suggestions for Authors
This article presents a multi-parameter method for assessing body composition in patients with colorectal cancer, which could be quite useful for clinicians in evaluating and managing such patients. However, there are several areas where further clarification from the authors is needed:
1. Although the study took into account factors such as gender, age, and weight, it may be necessary to further explore other potential confounding factors, such as tumor stage, treatment modalities (surgery, chemotherapy, radiotherapy, etc.), patients' baseline nutritional status, and lifestyle habits, all of which could influence body composition and prognosis.
2. The study did not include intervention studies based on the assessment results, such as nutritional support or physical activity interventions, and the impact of these interventions on improving body composition and clinical outcomes.
3. While body composition assessment tools were used, there was no mention of the use of other nutritional risk screening tools, such as the Nutritional Risk Screening 2002 (NRS-2002) or the Patient-Generated Subjective Global Assessment (PG-SGA). These tools could help identify patients who may require further nutritional support.
4. The article does not mention whether the assessment tools were cross-validated or their accuracy was verified in an independent sample.
Author Response
Comments 1: Although the study took into account factors such as gender, age, and weight, it may be necessary to further explore other potential confounding factors, such as tumor stage, treatment modalities (surgery, chemotherapy, radiotherapy, etc.), patients' baseline nutritional status, and lifestyle habits, all of which could influence body composition and prognosis.
Response 1: Thank you for highlighting these possible aspects that may act as confounding factors. Firstly, it should be clarified that the nutritional assessment carried out was performed at the time of diagnosis of the disease, so the patient had not received chemotherapy or radiotherapy. Secondly, the staging of the tumour is reflected in table 1, where it can be seen that 94.3% of the patients were in stage III or lower with an average BMI of 27 kg/m2 and only 37.1% met the criteria for malnutrition according to GLIM criteria. We agree that the study does not mention the interventions that are performed, so we provide this information in line number 120. This is an initial assessment, the results of these interventions and changes in body composition of these patients will be analysed and published in the near future.
Comments 2: The study did not include intervention studies based on the assessment results, such as nutritional support or physical activity interventions, and the impact of these interventions on improving body composition and clinical outcomes.
Response 2: We agree that the study does not mention the interventions that are performed, so we provide this information in line number 120. This is an initial assessment, the results of these interventions and changes in body composition of these patients will be analysed and published in the near future.
Comments 3: While body composition assessment tools were used, there was no mention of the use of other nutritional risk screening tools, such as the Nutritional Risk Screening 2002 (NRS-2002) or the Patient-Generated Subjective Global Assessment (PG-SGA). These tools could help identify patients who may require further nutritional support.
Response 3: In agreement, the classic screening tools for detecting malnutrition are still essential in clinical practice, which is why the GLIM criteria (line number 129) are used as a screening tool; but it has been shown that they are not useful for detecting cases of sarcopenic obesity, as could be the case in patients with colorectal cancer. According to the analysis we have been able to observe that 37.1% of the sample (99 patients) present malnutrition, I attach the data in table 1.
Comments 4: The article does not mention whether the assessment tools were cross-validated or their accuracy was verified in an independent sample.
Response 4: Indeed, the morphofunctional assessment tools are not validated individually, but as can be seen in the heatmaps in figure 2 and 3, there is a strong correlation between the morphological tools and those measuring functionality.

Reviewer 2 Report
Comments and Suggestions for Authors
Multiparametric approach to the color(r)ectal cancer phenotypes integrating morphofunctional assessment and computer tomography examine multiple methods to detect sarcopenia in CRC patients, as this is risk factor during treatment. One point is determining how well bedside methods work compared to CT, with the result that they seen to be adequate. The work seems through and can probably be published after some clarification.
There are so many abbreviations in the paper and derivative methods subclasses that it is confusing. In the Methods sections I recommend a Table with all of the methods/derivative methods be listed with abbreviations. In particular, I couldn't find "TC" and was guessing that this was an error, meaning CT? The biggest problem was trying to link the correlation coefficients from the text (lines 306-317) to fig. 2. Is the figure correct? Please, check this and Fig. 3
Comments on the Quality of English LanguageGenerally, OK. Misspelled colorectal in title.
Author Response
Comments 1: There are so many abbreviations in the paper and derivative methods subclasses that it is confusing. In the Methods sections I recommend a Table with all of the methods/derivative methods be listed with abbreviations. In particular, I couldn't find "TC" and was guessing that this was an error, meaning CT? The biggest problem was trying to link the correlation coefficients from the text (lines 306-317) to fig. 2. Is the figure correct? Please, check this and Fig. 3
Response 1: Thank you very much for the corrections, it is true that there must have been some error copying the numerical data from the correlation map of both figure 2 (lines 309-319) and figure 3 (lines 324-331). Corrections to the data can be found in the manuscript.

Reviewer 3 Report
Comments and Suggestions for Authors
This is a very interesting study. It highlights the value of using multiple body composition evaluation techniques, which provide a more thorough evaluation of the nutritional and functional status of patients with CCR. Overall, the manuscript was easy to follow. The title accurately reflects the results. The figures are representative of the data described in the results section, and the figure legends are sufficiently detailed. The discussion addresses how the results move the field forward and how these are correlated with existing literature in the field.
Please find below my comments about the manuscript:
- You may consider increasing the number of patient samples as they may not precisely represent a larger population of patients with postoperative CRC.
- Could you please clarify the data regarding age-dependent correlations? Are they scarce?
- Could you please clarify if the disease stage, gender, and geographic location may modify the prognostic effect of the integrated approach in CRC patients?
Author Response
Comments 1: You may consider increasing the number of patient samples as they may not precisely represent a larger population of patients with postoperative CRC.
Response 1: First of all, I would like to thank you for all the comments. It is true that it could be interesting to increase the sample, as CRC is a very prevalent cancer; we are also working on obtaining the data from the nutritional assessments and the TCs from the follow-up of these patients, so we hope to be able to publish the analysed data soon.
Comments 2: Could you please clarify the data regarding age-dependent correlations? Are they scarce?
Response 2: Referring to the linear regression data, we have observed that age is a factor affecting the quantification of muscle mass measured by TC, as are other factors such as weight and gender (table 6).
Comments 3: Could you please clarify if the disease stage, gender, and geographic location may modify the prognostic effect of the integrated approach in CRC patients?
Response 3: Finally, CT has been used to analyse body composition as an opportunistic technique. Patients diagnosed with CRC usually have a CT scan as a localisation or extension study before starting treatment. It is true that in countries with fewer economic resources it may not be available, so the prognosis of CRC patients may vary. In our study, for example, it was found that men had more visceral fat, which could lead to greater postoperative complications, but we still need to do more studies and, above all, increase the sample to prove this hypothesis.
Our aim with the comprehensive assessment of the patient's condition is to be able to make a more personalised treatment to improve the prognosis and nutritional status of patients with CRC.

Round 2
Reviewer 1 Report
Comments and Suggestions for Authors
Based on the author's response to the reviewer's comments and the revised manuscript, the paper has met the criteria for publication. The authors have provided ample information to explain their research methods and results, and have addressed my concerns. Therefore, I recommend that this paper be accepted for publication.
Author Response
Dear Reviewer,
We sincerely thank you for your positive comments and your recommendation to accept our manuscript for publication. We are pleased to know that the thorough review of the manuscript and our responses have successfully addressed your concerns and met the journal's quality criteria. Your contribution has been essential in enhancing the clarity and rigor of our work.
Thank you once again for your time and effort in reviewing our article.
Best regards
Reviewer 2 Report
Comments and Suggestions for Authors
The authors responded to most of my comments satisfactorily. On line 287, Title to Table 3. It still shows "by TC", which should read by CT as far as I can tell. This could be corrected in editing of the paper after acceptance.
Author Response
Dear Reviewer,
Thank you very much for your attentive review and for noting the remaining error in Table 3's title. We apologize for the oversight, and we will ensure it is corrected from "by TC" to "by CT" in the final editing process after acceptance.
We truly appreciate your careful reading and valuable feedback, which have contributed to improving the quality of our manuscript.
Best regards
